# Lossy compression with pretrained diffusion models

**Jeremy Vonderfecht & Feng Liu**
Department of Computer Science
Portland State University
{vonder2,fliu}@pdx.edu

## Abstract

We apply the DiffC algorithm (Theis et al., 2022) to Stable Diffusion 1.5, 2.1, XL, and Flux-dev, and demonstrate that these pretrained models are remarkably capable lossy image compressors. A principled algorithm for lossy compression using pretrained diffusion models has been understood since at least (Ho et al., 2020), but challenges in reverse-channel coding have prevented such algorithms from ever being fully implemented. We introduce simple workarounds that lead to the first complete implementation of DiffC, which is capable of compressing and decompressing images using Stable Diffusion in under 10 seconds. Despite requiring no additional training, our method is competitive with other state-of-the-art generative compression methods at low ultra-low bitrates.

## 1 Introduction

Today's AI labs are spending millions of dollars training state-of-the-art diffusion models. The original Stable Diffusion cost around $600,000 to train, while Stable Diffusion 3 is rumored to have cost around $10 million (Wikipedia, 2024). The sophistication and expense of these models are only likely to increase as generative modelling expands into the video domain.

While diffusion models are used primarily for prompt-based image generation, they are also highly powerful and flexible image priors. They provide many other affordances, such as easy multiplication with other distributions (i.e., posterior sampling), and log-likelihood estimation (Sohl-Dickstein et al., 2015). Researchers have discovered many remarkable applications of pretrained diffusion models, including image restoration (Chung et al., 2022), optical illusions (Geng et al., 2024b;a), and generative 3D modeling (Poole et al., 2022).

Ho et al. (2020) proposed an algorithm by which pretrained diffusion models can be leveraged for data compression. Using this algorithm, an image can be compressed to a number of bits approaching the model's negative log-likelihood estimate of the data. Leveraging this simple algorithm, could today's flagship diffusion models also serve as powerful image compressors?

Image compression remains a relatively underexplored application of diffusion models. Theis et al. (2022) produced the primary paper on this compression algorithm and christened it *DiffC*. Theis et al. (2022) cited the challenges associated with reverse-channel coding as a significant bottleneck to further adoption of the DiffC algorithm. For this reason, they do not offer a complete implementation of DiffC; they only evaluate its hypothetical performance, assuming a perfect solution to the reverse-channel coding problem. However, by introducing some simple workarounds, we find that reverse-channel coding induces only minor overhead costs in computation and bitrate. Our contributions are crucial for making DiffC a practical, usable algorithm instead of a hypothetical subject of study.

To the best of our knowledge, we are the first to fully implement the DiffC algorithm and the first to apply this algorithm to flagship open-source diffusion models, namely Stable Diffusion 1.5, 2, XL, and Flux-dev. We offer the first publicly available implementation of a DiffC compression protocol on GitHub[1]. Additionally, we propose a principled algorithm for optimizing the denoising schedule, which both improves our rate-distortion curve and reduces encoding time to less than

---

[1] https://github.com/jeremyiv/diffc

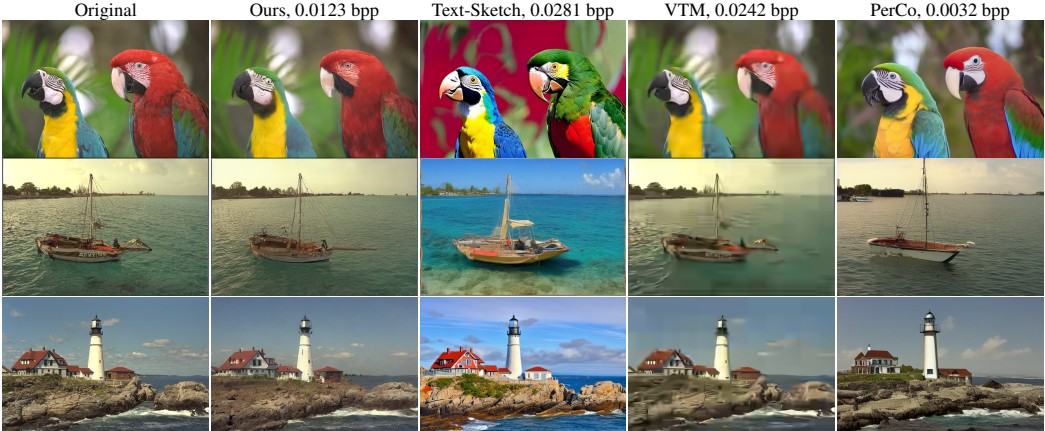

Figure 1: Kodak images compressed using our method on Stable Diffusion 1.5, Text-Sketch-PICS (Lei et al., 2023), VTM (vtm), and PerCo Careil et al. (2024). Text+Sketch/VTM/PerCo images are taken from Careil et al. (2024)

10 seconds with Stable Diffusion 1.5. Our algorithm works "zero-shot", requiring no additional training, and is naturally progressive, easily allowing images to be encoded to any desired bitrate. Remarkably, despite the fact that these diffusion models were not intended for image compression, we obtain competitive rate-distortion curves against other purpose-built state-of-the-art generative compression methods including HiFiC (Mentzer et al., 2020), MS-ILLM (Muckley et al., 2023), DiffEIC (Li et al., 2024), and PerCo (Careil et al., 2024).

## 2 BACKGROUND & RELATED WORK

Our work is founded on the DiffC algorithm introduced by Theis et al. (2022). We contribute several practical advancements to implementing DiffC. While Theis et al. applied DiffC to a variant of VDM (Kingma et al., 2021) trained on 16x16 ImageNet patches, and only analyzed theoretical compression rate lower bounds, we adapt the algorithm to run efficiently on Stable Diffusion and Flux.

### 2.1 LOSSY COMPRESSION WITH GENERATIVE MODELS

Methods leveraging generative models have managed to compress images to unprecedented bitrates while maintaining high perceptual quality. The dominant paradigm in this field encodes images by quantizing the latent representations of variational autoencoders. For example, see Minnen et al. (2018), HiFiC (Mentzer et al., 2020), MS-ILLM (Muckley et al., 2023), and GLC Jia et al. (2024). Since diffusion models can be regarded as a special case of VAEs (Kingma et al., 2021), DiffC can also be considered a member of this class of compression techniques. But while other VAE-based approaches build sophisticated entropy models for their latent representations, DiffC has an elegant and principled entropy model "built-in" to the algorithm.

Diffusion models represent a powerful and flexible class of generative models for images. Naturally, researchers have sought to leverage them for image compression. Elata et al. (2024) propose Posterior-Sampling based Compression (PSC), which encodes an image using quantized linear measurements of that image, and uses those measurements for posterior sampling from a diffusion model for lossy reconstruction. Like DiffC, PSC can be applied to pretrained diffusion models with no additional training. However, it is also more computationally intensive, as it requires posterior sampling from a diffusion model many times to encode a single image. Xu et al. (2024) use DDPM inversion to generate reconstructions with the same compressed encoding as the original image.

The dominant paradigm in diffusion-based compression is to use a conditional diffusion model, and to encode the target image as a compressed representation of the conditioning information. Text+Sketch (Lei et al., 2023) uses ControlNet (Zhang et al., 2023) to reconstruct an image from

| **Algorithm 1** Sending $\mathbf{x}_0$ (Ho et al., 2020) | **Algorithm 2** Receiving |
|---|---|
| 1: Send $\mathbf{x}_T \sim q(\mathbf{x}_T|\mathbf{x}_0)$ using $p(\mathbf{x}_T)$ | 1: Receive $\mathbf{x}_T$ using $p(\mathbf{x}_T)$ |
| 2: **for** $t = T - 1, \ldots, 2, 1$ **do** | 2: **for** $t = T - 1, \ldots, 1, 0$ **do** |
| 3:     Send $\mathbf{x}_t \sim q(\mathbf{x}_t|\mathbf{x}_{t+1}, \mathbf{x}_0)$ using $p_\theta(\mathbf{x}_t|\mathbf{x}_{t+1})$ | 3:     Receive $\mathbf{x}_t$ using $p_\theta(\mathbf{x}_t|\mathbf{x}_{t+1})$ |
| 4: **end for** | 4: **end for** |
| 5: Send $\mathbf{x}_0$ using $p_\theta(\mathbf{x}_0|\mathbf{x}_1)$ | 5: **return** $\mathbf{x}_0$ |

a caption and a highly compressible Canny edge detection-based "sketch" of the original image. PerCo (Careil et al., 2024) and CDC (Yang & Mandt, 2024) train conditional diffusion models to reconstruct images from quantized embeddings. DiffC is more general, in that it can be applied to any diffusion model, with or without additional conditioning. In fact, DiffC can easily be combined with these conditional diffusion approaches, to more precisely guide the denoising process towards the ground truth. For example, our prompt-conditioned method could be considered an extension of Text+Sketch's PIC method. While we do not combine DiffC with other conditional diffusion approaches, like PerCo or CDC, we regard this as an interesting future research direction.

## 2.2 Diffusion Denoising Probabilistic Models

This section provides a brief review of the DDPM forward and reverse processes. For a more comprehensive introduction to these concepts, see Sohl-Dickstein et al. (2015) and Ho et al. (2020).

DDPMs learn to reverse a stochastic *forward process* $q$. Let $q(\mathbf{x}_0, \mathbf{x}_1, \ldots \mathbf{x}_T)$ be a joint probability distribution over samples $\mathbf{x}_t \in \mathbb{R}^d$. Let the marginal distribution $q(\mathbf{x}_0)$ be the training data distribution. Subsequent samples $x_1, \ldots, x_T$ are sampled from a random walk starting at $\mathbf{x}_0$ and evolving according to:

$$q(\mathbf{x}_t|\mathbf{x}_{t-1}) := \mathcal{N}(\mathbf{x}_t; \sqrt{1 - \beta_t}\mathbf{x}_{t-1}, \beta_t\mathbf{I})$$

where $\beta_t$ is the "beta schedule" which determines how much random noise to add at each step. The beta schedule is chosen so that $q(\mathbf{x}_T) \approx \mathcal{N}(0, \mathbf{I})$. This defines a random process by which samples from our training distribution $q(\mathbf{x}_0)$ are gradually transformed into samples from a standard normal distribution $q(\mathbf{x}_T)$.

To generate samples from the training distribution $q(\mathbf{x}_0)$, DDPMs construct a *reverse process*; starting from a Gaussian noise sample $\mathbf{x}_T$ and then generating a sequence of successively denoised samples $\mathbf{x}_{T-1}, \ldots \mathbf{x}_0$. To do so, they must approximate $q(\mathbf{x}_{t-1}|\mathbf{x}_t)$. Happily, as $\beta_t$ becomes small, $q(\mathbf{x}_{t-1}|\mathbf{x}_t)$ becomes Gaussian. DDPMs parameterize this distribution as a neural network $p_\theta(\mathbf{x}_{t-1}|\mathbf{x}_t)$, and train the network to minimize the KL divergence $D_{KL}(q(\mathbf{x}_{t-1}|\mathbf{x}_t)||p_\theta(\mathbf{x}_{t-1}|\mathbf{x}_t))$.

Importantly, this training objective is equivalent to optimizing the variational bound on the log likelihood: $\mathcal{L}(\mathbf{x}_0) \leq \mathbb{E}_q[\log p_\theta(\mathbf{x}_0)]$, which is the exact objective we want for data compression. In the next section, we will explain how to communicate a sample $\mathbf{x}_0$ using close to $-\mathcal{L}(\mathbf{x}_0)$ bits.

## 2.3 DDPM-Based Compression

DDPM-based image compression works similarly to DDPM-based sampling: we start from a random Gaussian sample $\mathbf{x}_T$ and progressively denoise it to produce a sequence of samples $\mathbf{x}_{T-1}, \mathbf{x}_{T-2}, \ldots, \mathbf{x}_0$. To sample a random $\mathbf{x}_0$, we draw these successive denoised samples from $p_\theta(\mathbf{x}_{t-1}|\mathbf{x}_t) \approx q(\mathbf{x}_{t-1}|\mathbf{x}_t)$. But to compress a specific target image $\mathbf{x}_0$, we instead wish to draw our denoised samples from $q(\mathbf{x}_{t-1}|\mathbf{x}_t, \mathbf{x}_0)$. Fortunately, this is a tractable Gaussian distribution.

At a high level, our compression algorithm sends the minimal amount of information needed to "steer" the denoising process to sample from $q(\mathbf{x}_{t-1}|\mathbf{x}_t, \mathbf{x}_0)$ instead of $p_\theta(\mathbf{x}_{t-1}|\mathbf{x}_t)$. The receiver doesn't have access to $\mathbf{x}_0$, but can compute $p_\theta(\mathbf{x}_{t-1}|\mathbf{x}_t)$, which is conveniently optimized to minimize the expected value of $D_{KL}(q(\mathbf{x}_{t-1}|\mathbf{x}_t, \mathbf{x}_0)||p_\theta(\mathbf{x}_{t-1}|\mathbf{x}_t))$. Using *reverse-channel coding* (RCC), it is possible to send a random sample from $q$ using a shared distribution $p$ using close to $D_{KL}(q||p)$ bits. Algorithms 1 and 2 detail the full compression protocol.

While Algorithms 1 and 2 communicate $\mathbf{x}_0$ losslessly, they permit a simple modification for lossy compression: simply stop after any time $t$ to communicate $\mathbf{x}_t$, which is a noisy version of $\mathbf{x}_0$. Then use a denoising algorithm to reconstruct $\hat{\mathbf{x}}_0 \approx \mathbf{x}_0$. Theis et al. (2022) propose using flow-based DDPM sampling to perform the final denoising step, as this yields naturalistic images with a higher PSNR than standard ancestral sampling. For a more detailed explanation of the DiffC algorithm, see Theis et al. (2022).

## 2.4 REVERSE-CHANNEL CODING

The central operation in Algorithms 1 and 2 is to "send $x \sim q(x)$ using $p(x)$". In other words, we wish to communicate a random sample $x$ from a distribution $q(x)$ using shared distribution $p(x)$, using roughly $D_{kl}(q||p)$ bits, and a shared random generator. This is a basic problem in information theory called *reverse-channel coding* (RCC). We use the *Poisson Functional Representation* (PFR) algorithm (4) from Theis & Ahmed (2022) for reverse channel coding, as this algorithm achieves perfect sampling from the target distribution in only slightly more than $D_{kl}(q||p)$ bits.

## 3 METHOD

As mentioned in Section 2.3, the DiffC algorithm operates in two stages:

1. First, communicate a noisy sample of the data $x_t$ using Algorithms 1 and 2.

2. Then, denoise this sample using probabiltiy-flow-based DDPM denoising.

The challenge of implementing this algorithm lies in making RCC efficient. In this section, we will describe this challenge and our solution to it.

In a theoretically idealized version of DiffC, communicating $\mathbf{x} \sim q$ using $p$ would be possible using exactly $D_{KL}(q||p)$ bits. Under these idealised conditions, communicating a noisy sample $\mathbf{x}_t$ requires

$$\sum_{i=T}^{t} D_{KL}(q(\mathbf{x}_{i-1}|\mathbf{x}_i)||p_\theta(\mathbf{x}_{i-1}|\mathbf{x}_i)) = -\mathcal{L}(\mathbf{x}_t) \tag{1}$$

bits of information. This is the appeal of using large pretrained diffusion models for compression: as the models increase in scale, we expect $\mathcal{L}(\mathbf{x}_t)$ to asymptotically approach the ideal "true" log-likelihood of $\mathbf{x}_t$.

Unfortunately, there is no known algorithm for doing this efficiently. Instead, one of the best-known algorithms for RCC is PFR (4), which allows us to us to send $\mathbf{x} \sim q$ using $p$ using roughly

$$D_{KL}(q||p) + \log(D_{KL}(q||p)) + 5$$

bits. As $D_{KL}(q||p)$ becomes large, this bitrate overhead becomes neglegible. Unfortunately, PFR's time complexity is exponential in $D_{KL}(q||p)$. So when $D_{KL}(q||p)$ is too large, the encoding time is impractical. But when $D_{KL}(q||p)$ is too small, PFR is bitrate-inefficient. So there is a "sweet spot" of $D_{KL}(q||p)$ values for which both the encoding time and bitrate overheads are tolerable. With our fast CUDA implementation of PFR, we can RCC 16 bits at a time with negligible computational overhead.

In practice, $D_{KL}(q(\mathbf{x}_{t-1}|\mathbf{x}_t,\mathbf{x}_0)||p_\theta(\mathbf{x}_{t-1}|\mathbf{x}_t))$ is impractically small when $t$ is close to $T$ (as little as 0.25 bits per step), and impractically large when $t$ is close to 0 (thousands of bits per step). Fortunately, there are workarounds for both cases:

- When $D_{KL}$ per denoising step is too small, we can take larger steps. For example, instead of sending $x_{999} \sim q(\mathbf{x}_{999}|\mathbf{x}_{1000}, \mathbf{x}_0)$ using $p_\theta(\mathbf{x}_{999}|\mathbf{x}_{1000})$), We can send $x_{990} \sim q(\mathbf{x}_{990}|\mathbf{x}_{1000}, \mathbf{x}_0)$ using $p_\theta(\mathbf{x}_{990}|\mathbf{x}_{1000})$). Skipping denoising steps during DDPM sampling is already common practice for image generation, and DiffC faces similar compute/likelihood tradeoffs. Section 4.3 examines this tradeoff in more detail.

- When $D_{KL}$ is too large, we can randomly split up the dimensions of $q$ and $p$ into $n$ statistically independent distributions, $q = q_1 \cdot q_2 \ldots q_n$ and $p = p_1 \cdot p_2 \ldots p_n$. We select $n$ so that $D_{KL}(q_i \| p_i)$ is near our sweet spot of 16 bits for all $i \in [1, n]$. This is possible because $q$ and $p$ are high-dimensional anisotropic Gaussian distributions, so each dimension can be sampled (and RCC'd) independently. And as long as the dimension of $\mathbf{x}$ is much greater than $n$, the law of large numbers ensures that:

$$D_{\mathrm{KL}}(q_i \parallel p_i) \approx \frac{1}{n} D_{\mathrm{KL}}(q \parallel p) \quad \forall i \in [1, n]$$

With careful choices of hyperparameters, we have managed to produce a fast implementation of DiffC for Stable Diffusion that incurs a $< 30\%$ bitrate overhead to achieve the same PSNR as the idealized method (see Figure 2).

## 3.1 CUDA-BASED REVERSE-CHANNEL CODING

Unoptimized implementations of reverse-channel coding may still be prohibitively slow at 16 bits per chunk. The TensorFlow implementation of reverse-channel coding from MIRACLE (Havasi et al., 2019) takes about 140 ms per 16-bit chunk of our data with an A40 GPU. For a 1 kilobyte image encoding using our Stable Diffusion 1.5 compression method, this implementation of RCC takes 80% of the encoding time; requiring about 80 seconds per image to produce an encoding.

The key issue is that standard PyTorch/Tensorflow implementations of reverse-channel coding must materialize large arrays of random samples in order to take advantage of GPU paralellism. Our custom CUDA kernel can avoid these memory requirements. For more details, see Appendix A.5.

Our implementation of reverse-channel coding is approximately 64x faster than what we could achieve with PyTorch or Tensorflow alone. Encoding a 16-bit chunk takes less than 3 milliseconds; RCC becomes a neglegible contribution to the overall runtime of this compression algorithm. This allows us to encode an image in less than 10 seconds.

## 3.2 GREEDY OPTIMIZATION OF DIFFC'S HYPERPARAMETERS

The DiffC algorithm communicates a sequence of successively denoised samples of the target image. These noisy samples have a convenient property: each $\mathbf{x}_t$ we send is statistically indistinguishable from a random sample of $q(\mathbf{x}_t | \mathbf{x}_0)$. As long as the samples we communicate obey this property, any choices we make about how to obtain $\mathbf{x}_t$ have no further side effects. Each subsequent denoising step can be greedily optimized to obtain $x_t$ using as little information as possible. We use this fact to optimize the timestep schedule of our denoising process as follows:

To communicate $\mathbf{x}_t$ efficiently, we wish to choose a timestep schedule $S = [T, t_i, t_j, \ldots, t_{\text{final}}]$ which minimizes the expected number of bits needed to communicate $\mathbf{x}_{t_{\text{final}}}$. Let $C(i, j)$ be the coding "cost" (e.g. required bits) of sending $\mathbf{x}_j$ given that $\mathbf{x}_i$ has already been received. For example, using Algorithm 4, $C(i, j) = I + \log(I) + 5$, where $I = D_{KL}(q(\mathbf{x}_j | \mathbf{x}_i, \mathbf{x}_0) \| p(\mathbf{x}_j | \mathbf{x}_i))$. So the cost of an entire timestep schedule S is:

$$C(S) = \sum_{i=1}^{|S|-1} (C(S_i, S_{i+1}))$$

Fortunately, $C(i, j)$ does not depend on how $x_i$ was communicated. By the nature of our compression algorithm, each $\mathbf{x}_i$ we send is statistically indistinguishable from a random sample of $q(\mathbf{x}_i | \mathbf{x}_0)$. Therefore, optimizing the hyperparameters of Algorithm 1 is a problem with *optimal substructure*. Let $C_{\text{opt}}[i, j]$ be the minimum cost (however defined) of sending a noisy sample $\mathbf{x}_{t_j} \sim q(\mathbf{x}_{t_j} | \mathbf{x}_{t_i}, \mathbf{x}_0)$ given a shared sample $\mathbf{x}_{t_i} \sim q(\mathbf{x}_{t_i} | \mathbf{x}_0)$. Then:

$$C_{\text{opt}}[i, j] = min_k C[i, k]_{\text{opt}} + C[k, j]_{\text{opt}}$$

Therefore, we can use a greedy algorithm to find the timestep schedule which will require the minimum expected number of bits to communicate $\mathbf{x}_t$. We optimize our timestep schedule for a set of

---

**Algorithm 3** Optimal DiffC Timestep Schedule

---

**Require:**
    $p$: reverse process specified by the diffusion model
    $q$: diffusion forward process
    $X$: dataset of images to encode (e.g. Kodak 24)
    $C(D_{\mathrm{KL}})$: coding cost function, where $C(D_{\mathrm{KL}}(q||p))$ is the cost of sending $\mathbf{x} \sim q$ using $p$
    $t_{\mathrm{final}}$: timestep of the last noisy sample to communicate
**Ensure:** $S$: Minimum average cost timestep schedule $\{T, ..., t_{\mathrm{final}}\}$
 1: $D[\mathbf{x}, i, j] \leftarrow \infty$ for all $\mathbf{x}, i, j$
 2: **for** $\mathbf{x}_0 \in X$ **do**
 3:     Sample $\mathbf{x}_T \sim q(\mathbf{x}_T|\mathbf{x}_0)$ using $p(\mathbf{x}_T)$
 4:     **for** $i = T, T-1, \ldots, t_{\mathrm{final}} + 1$ **do**
 5:         **for** $j = i-1, i-2, \ldots, t_{\mathrm{final}}$ **do**
 6:             $D[\mathbf{x}, i, j] \leftarrow D_{\mathrm{KL}}(q(\mathbf{x}_j|\mathbf{x}_i, \mathbf{x}_0)||p(\mathbf{x}_j|\mathbf{x}_i))$
 7:         **end for**
 8:     **end for**
 9: **end for**
10: $C[i, j] \leftarrow \mathrm{mean}(C(D[\mathbf{x}, i, j])$ for $\mathbf{x} \in X)$
11: $S \leftarrow \mathrm{ShortestPath}(C, T, t_{\mathrm{final}})$     ▷ Find shortest path from $T$ to $t_{\mathrm{final}}$ in adjacency matrix $C$
12: **return** $S$

---

images $X$: for the Kodak dataset, $X$ consists of all 24 images. For Div2K we choose a random sample of 30 images. See Algorithm 3:

Note that Algorithm 3 requires $|X|T$ forward passes through the diffusion model, as we only need to perform one forward pass for each new $\mathbf{x}_i$. Given the model's noise prediction $\epsilon_i$ there is a simple closed-form equation to compute $p(\mathbf{x}_j|\mathbf{x}_i)$ for any $j$.

### 3.3   DENOISING $\mathbf{x}_t$

To evaluate DiffC at different bitrates, we generate lossy reconstructions from noisy samples $\mathbf{x}_t$ at $t \in [900, 800, \ldots, 100, 90, \ldots 10]$.

Once we have communicated the noisy sample $\mathbf{x}_t$, we have several options as to how to denoise it. Theis et al. (2022) proposed "DiffC-A" (ancestral sampling) and "DiffC-F" (probability flow based sampling). Ancestral sampling simply continues the stochastic DDPM denoising process to arrive at a fully denoised image. DiffC-F denoises $\mathbf{x}_t$ by following the probability flow ODE. As far as we can tell, DiffC-F is strictly superior, in that it requires fewer inference steps, and results in lower distortion than DiffC-A. Therefore, we adopt DiffC-F as our main denoising method. For simplicity we follow the probability flow with a standard 50-step DDIM scheduler (Song et al., 2020).

In Appendix A.2, we consider a third denoising method, which simply predicts $\hat{\mathbf{x}}_0$ from $\mathbf{x}_t$ with a single forward-pass through the diffusion model. While there are a-priori reasons for this method to seem interesting, we did not find it valuable in practice.

### 3.4   ADAPTING DIFFC FOR FLUX

DiffC was originally formulated for Diffusion Denoising Probabilistic Models (DDPMs) (Ho et al., 2020), but Flux-dev is trained with Optimal Transport Flow Matching (OT flow) (Lipman et al., 2022). Fortunately, there is a simple transformation that converts OT flow probability paths into DDPM probability paths, allowing us to treat Flux-dev exactly like a DDPM. For details, see Appendix A.3.

### 3.5   FURTHER IMPLEMENTATION DETAILS

**Fixed Bitrate Per Step:** To efficiently communicate a random sample from q using p, the sender and receiver must pre-establish $D_{KL}(q||p)$. But $D_{KL}(q||p)$ varies with the contents of the encoded image. To establish a complete compression protocol, the sender and receiver must establish the

Table 1: Parameter count and average encoding and decoding times (in seconds) on an A40 GPU for Kodak/Div2k-1024 images. DiffC encoding/decoding times depend on the number of denoising steps taken, which can vary from roughly 10-150 for encoding, and from 50-150 for decoding, depending on the bitrate and reconstruction quality. Encoding is slightly slower per step than decoding due to the asymmetry of reverse-channel coding.

| Method | # Params | Encoding (s) | Decoding (s) |
|---|---|---|---|
| MS-ILLM (Kodak) | 181M | 1.6 | 2.9 |
| MS-ILLM (Div2K) | 181M | 2.3 | 4.8 |
| HiFiC (Div2k) | 181M | 1.5 | 3.8 |
| PerCo (Kodak) | 950M | 0.1 | 3.6 |
| DiffEIC (Kodak) | 950M | 0.2 | 6.6 |
| DiffC (SD 1.5) (Kodak) | 943M | 0.6–9.4 | 2.9–8.5 |
| DiffC (SD 2.1) (Kodak) | 950M | 0.6–8.8 | 2.8–8.2 |
| DiffC (SDXL) (Div2k) | 2.6B | 3.2–46.2 | 15.3–36.6 |
| DiffC (Flux) (Div2k) | 12B | 21.0–321.4 | 98.5–215.0 |

KL divergence of the distribution being reverse-channel coded in each step. In principle, you could develop an entropy coding model for this side-information. But in practice, we have found that just hard-coding a sequence of expected $D_{KL}$ values into the protocol based on their averages does not affect the performance of our method too much. In fact, we were surprised at the extent to which the R-D curves are robust to variations in the protocol's $D_{KL}$ and timestep schedules. See Appendix A.6 for further results.

**SDXL Base vs. Refiner:** SDXL consists of two models, a base model and a "refiner", which is specialized for the last 200 of the 1,000 denoising steps (Podell et al., 2023). We found no meaningful difference between the compression rates achieved by the base and refiner models of SDXL in the high signal-to-noise ratio domain. However, using the refiner during reconstruction leads to a modest improvement in CLIP and quality scores. See Figure 2 for comparison. Therefore we elected to use the refiner for the last 20% of the denoising process as Podell et al. (2023) recommends.

## 4 RESULTS

### 4.1 LATENT DIFFUSION LIMITS FIDELITY

Algorithms 1 and 2 allow for exact communication of a sample $x_0$. However, with latent diffusion models, $\mathbf{x}_0$ does not represent the original image, but its latent representation. Therefore, the fidelity of our compression algorithm is limited by the reconstruction fidelity of these models' variational autoencoders. On the Kodak dataset, SD1.5/2.1's variational autoencoder achieves an average PSNR of 25.7 dB, while SDXL's VAE obtains 28.8 dB. Therefore, Stable Diffusion-based compression is primarily useful in the ultra-low bitrate regime, where average PSNRs under 25 dB are competitive with the state of the art. However, Flux's VAE is much higher fidelity, obtaining 32.4 dB PSNR on the Kodak dataset. Therefore Flux is capable of much higher fidelity compression, and remains competitive with HiFiC at 0.46 bits per pixel. Figure 2 displays distortion results in reference to this VAE bound.

### 4.2 RATE-DISTORTION CURVES

Here, we evaluate the rate-distortion curves for our DiffC algorithm under a variety of settings. Figure 2 shows our primary results, evaluating the performance of our methods on Kodak and DIV2K images with Stable Diffusion 1.5, 2.1, XL, and Flux-dev. We compare against HiFiC (Mentzer et al., 2020), MS-ILLM (Muckley et al., 2023), DiffEIC (Li et al., 2024), PerCo (Careil et al., 2024), and VVC-intra (Wieckowski et al.). Distortion metrics used are PSNR, LPIPS (Zhang et al., 2018), and CLIP scores (Hessel et al., 2021). Q-Align (Wu et al., 2023) is a no-reference image quality metric which we believe is more meaningful than Frechet Inception Distance (Heusel et al., 2017) given our relatively small datasets. Figure 3 gives a visual comparison of DiffC to other generative compression methods.

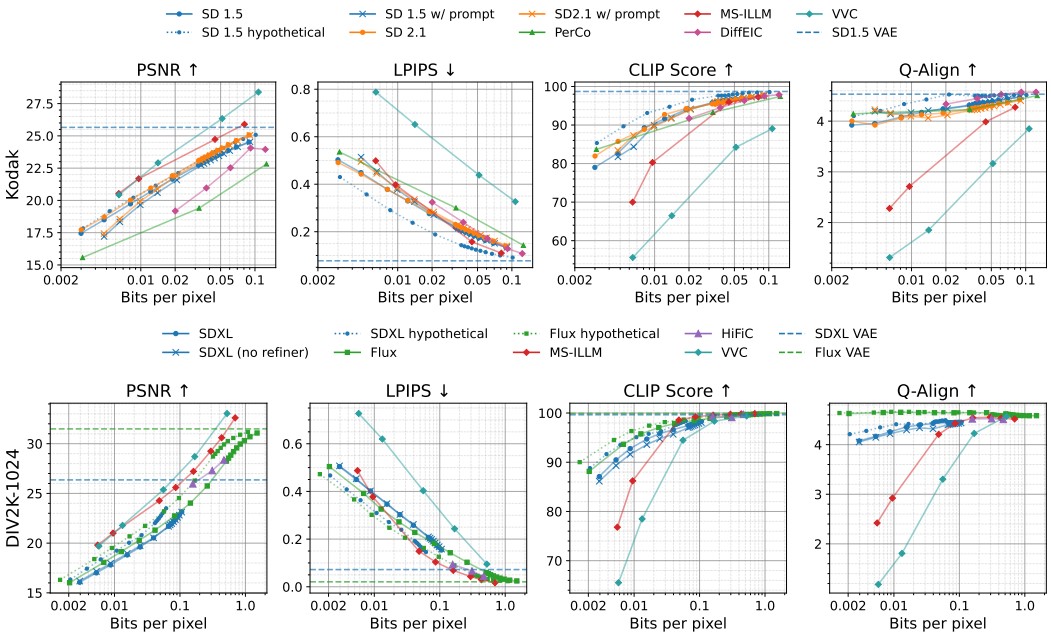

Figure 2: Rate-distortion curves for generative compression methods across three sets of images. "Hypothetical" refers hypothetically-optimal RD curves assuming ideal reverse-channel coding. Best viewed zoomed in.

From these and other data we make the following qualitative observations:

- The differences between the RD curves of different flagship diffusion models is relatively modest: SD1.5 and 2.1 have comparable performance on the Kodak dataset, while SDXL and Flux have comparable performance on the Div2K dataset. The primary advantage of Flux appears to be that it is capable of much higher bitrate compression, and higher reconstruction quality. We speculate that larger and more sophisticated diffusion models may be asymptotically approaching the "true" log-likelihood of the data, so that larger diffusion models will yield in increasingly small bitrate improvements.

- Hypothetically ideal reverse-channel coding yields better R-D curves than our practical implementation. Improvements to our reverse-channel coding scheme might help realize these models' full compressive potential.

- We do not find prompts to be worth their weight in bits. We tried both BLIP captions (Li et al., 2022) and Hard Prompts Made Easy (Wen et al., 2024). We found that a prompt guidance scale near 1 was optimal for communicating the noisy latent, and denoising with a guidance scale around 5 was optimal for maximizing CLIP scores. For both methods, we found that you would be better off just allocating those bits directly to DiffC without prompt conditioning. While Figure 2 only shows the effect of prompt conditioning for Stable Diffusion 1.5, the effect for 2.1 was almost identical. For Flux and SDXL, prompt conditioning offered no improvement to distortion metrics, but did significantly improve reconstruction quality at lower bitrates (see Appendix Figure 7).

- DiffC achieves competitive rate-distortion curves against other state-of-the-art generative compression methods, and has a place of the rate/distortion/perception Pareto frontier. It obtains lower distortion than PerCo or DiffEIC, and it achieves higher perceptual quality than HiFiC and MS-ILLM. Of course its other advantage is that DiffC works zero-shot and is naturally progressive, while all other methods compared here require specialized training for each new bitrate.

### 4.3 RATE-DISTORTION VS. NUMBER OF INFERENCE STEPS

We analyze the tradeoff between the number of inference steps in our denoising schedule vs. the rate-distortion curve. To determine the best denoising timestep schedule, we use a greedy optimization

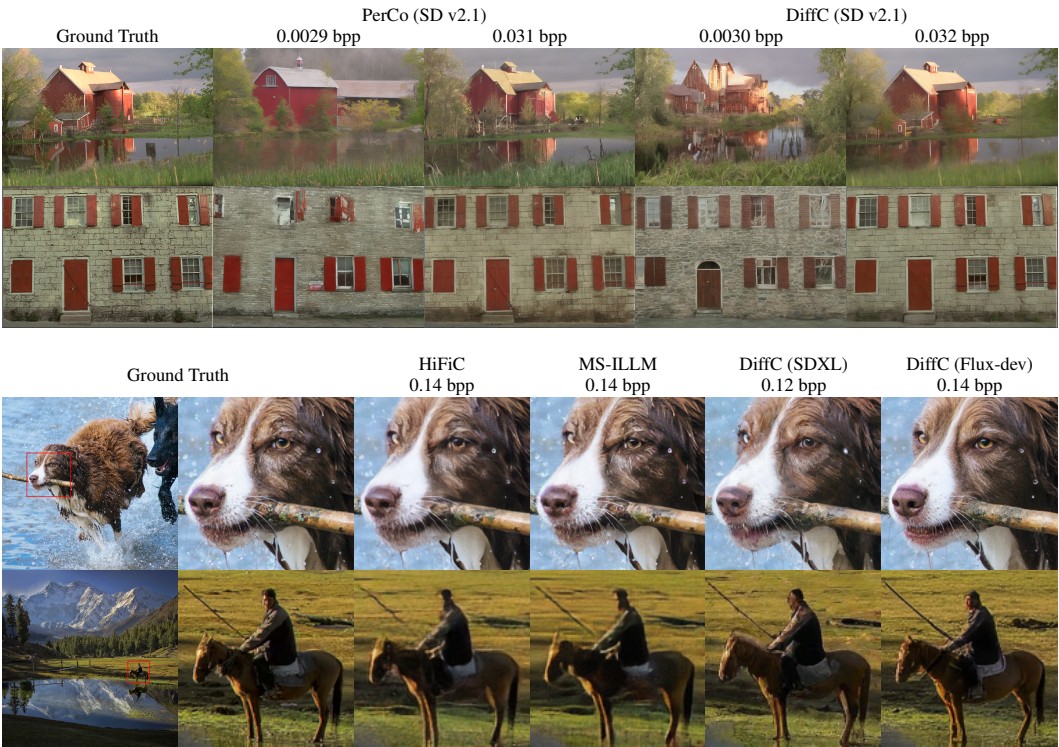

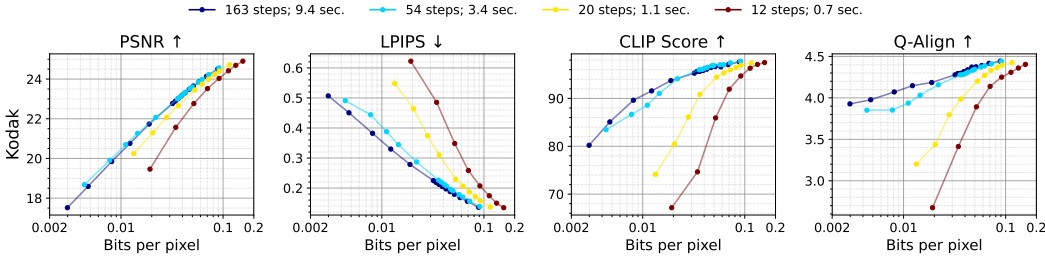

Figure 3: Visual comparison of generative compression methods.

Figure 4: RD curves on Kodak dataset vs. number of RCC steps. Legend shows encoding time per image in seconds.

technique that minimizes the total number of bits needed to reach a target timestep $t$. For more details, see Algorithm 3. Figure 4 shows rate-distortion curves for SD1.5 on the Kodak dataset using between 12 and 163 total steps. (We do not test more steps, since our greedy optimization technique indicates that that 163 steps is bitrate-optimal, given the reverse-channel coding overhead.) Notice that the number of steps can be reduced to 54 with only minor degradation in quality. Table 1 shows the number of model parameters and the encoding/decoding times for each compression method.

## 5 CONCLUSION

By solving several challenges surrounding reverse-channel coding, we introduce the first practical implementation of DiffC. We evaluate our implementation using Stable Diffusion 1.5, 2.1, XL, and Flux-dev. Our method is competitive with other generative compression methods on rate/distortion/perception curves. Our method outperforms PerCo and DiffEIC on rate-distortion curves, and exceeds HiFiC and MS-ILLM in perceptual image quality. Compared to other generative compression methods, which typically must be trained for each new bitrate, DiffC can be applied zero-shot

to pretrained diffusion models to compress images to any bitrate. However, the slowness of diffusion inference, and the large number of required denoising steps, remains a significant bottleneck.

## 6 FUTURE WORK

We believe there is a wide frontier of future work to be done on DiffC applications. We will conclude by highlighting four potential directions: inference speed, Rectified Flow models, flexible image sizes, and conditional diffusion models.

- **Inference time:** Despite our alleviation of the RCC bottleneck, these compression algorithms are still quite slow. The DiffC algorithm requires at least tens of inference steps, and large models such as Flux can require multiple seconds per inference step, even on powerful GPUs. For Flux, this can result in an encoding time of more than five minutes per image. However, we remain optimistic that there are many potential avenues to faster compression. As one example, diffusion activation caching (Moura et al., 2019; Ma et al., 2024) could eliminate redundant computation between denoising steps.

- **Rectified Flow models:** DiffC is designed for DDPM-based diffusion models, and can be easily adapted to Optimal Transport Flow models (see Appendix A.3). (Lipman et al., 2022). However, state-of-the-art diffusion models such as Stable Diffusion 3 (Esser et al.) and Flux are moving towards rectified flow (Liu et al., 2022). It is not obvious how to adapt DiffC to leverage rectified flow diffusion models. But for this class of compression algorithms to remain relevant to the most state-of-the-art diffusion models, this limitation must somehow be overcome.

- **Flexible image sizes:** Our compression method inherits Stable Diffusion's image size limitations: performance degrades quickly for image sizes outside the training distribution. Extending the capabilities of pretrained diffusion models to image sizes outside their training distribution is an active area of research (Haji-Ali et al., 2024), and progress in generation may be applicable to compression as well.

- **Conditional Diffusion Models:** Another clear avenue for future work is to combine DiffC with information to condition the diffusion process. Following Lei et al. (2023), we have tried conditioning on prompts generated by Hard Prompts Made Easy (Wen et al., 2024). But this could be taken much further. For example, one could try using an entropy model to directly compress an image's CLIP embedding. As mentioned in Section 2.1, it could also be interesting to apply DiffC on top of other conditional diffusion compression methods, such as PerCo (Careil et al., 2024) or CDC (Yang & Mandt, 2024).

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

| Metric | SRCC ↑ |
|--------|--------|
| PSNR | 0.406 |
| SD1.5 VAE | 0.574 |
| LPIPS | 0.584 |
| SDXL VAE | 0.600 |
| Flux VAE | 0.656 |

Table 2: SRCC against user mean opinion scores on the PIPAL dataset.

## A APPENDIX

### A.1 THE POISSON FUNCTIONAL REPRESENTATION (PFR) REVERSE-CHANNEL CODING ALGORITHM.

---
**Algorithm 4** PFR, Theis & Ahmed (2022)

---
**Require:** $p, q, w_{\min}$
1: $t, n, s^* \leftarrow 0, 1, \infty$

2: **repeat**
3:     $z \leftarrow \mathtt{simulate}(n, p)$          ▷ Candidate generation
4:     $t \leftarrow t + \mathtt{expon}(n, 1)$          ▷ Poisson process
5:     $s \leftarrow t \cdot p(z)/q(z)$          ▷ Candidate's score

6:     **if** $s < s^*$ **then**          ▷ Accept/reject candidate
7:         $s^*, n^* \leftarrow s, n$
8:     **end if**

9:     $n \leftarrow n + 1$
10: **until** $s^* \leq t \cdot w_{\min}$

11: **return** $n^*$

---

In this algorithm, $\mathtt{simulate}$ is a shared pseudorandom generator that takes a random seed $n$ and a distribution $p$ and draws a pseudorandom sample $z \sim p$.

### A.2 DIFFC-MSE

We originally considered an additional denoising method: "DiffC-MSE". This is a 1-step denoiser which simply predicts $\hat{x}_0$ from $x_t$ with a single forward-pass through the diffusion model. $\hat{x}_0$ minimizes the expected distortion against the true $x_0$, as measured by euclidean distance in the diffusion model's latent space.

This is an interesting property because we find that euclidean distance in the latent spaces of SD 1.5/2.1, XL, and Flux correlate surprisingly well with perceptual similarity. We used the PIPAL dataset (Jinjin et al., 2020) to evaluate the correlation of VAE latent distance with users' perceptual similarity rankings using Spearman's rank correlation coefficient (SRCC). Table 2 shows the results.

Therefore, it stands to reason that DiffC-MSE may yield reconstructions with lower perceptual distortion than either DiffC-A or DiffC-F, at the expense of realism. However, in our experiments, we found that "DiffC-MSE" was almost universally inferior to DiffC-F along all distortion metrics. Hence its relegation to the appendix.

### A.3 TRANSFORMING OPTIMAL TRANSPORT FLOW INTO STANDARD GAUSSIAN DIFFUSION

Flux-dev is trained with Optimal Transport Flow (OT flow) (Lipman et al., 2022). To apply the DiffC algorithm, we must first transform it into a standard DDPM. Fortunately, DDPM and OT flow probability paths are equivalent under a simple transformation.

Both DDPM and OT flow objectives can be formulated as trying to predict the expected original signal $\hat{\mathbf{x}}_0$ given a noisy sample $\mathbf{x}_t = s_t\mathbf{x}_0 + n_t\boldsymbol{\epsilon}$, where $\boldsymbol{\epsilon} \sim \mathcal{N}(\mathbf{0}, \mathbf{I})$.

For OT flow models:

$$\mathbf{x}_t = (1 - \sigma_t)\mathbf{x}_0 + \sigma_t\boldsymbol{\epsilon}$$

For DDPMs:

$$\mathbf{x}_t = \sqrt{\bar{\alpha}_t}\mathbf{x}_0 + \sqrt{1 - \bar{\alpha}_t}\boldsymbol{\epsilon}$$

where $\alpha_t = 1 - \beta_t$ and $\bar{\alpha}_t = \prod_{s=1}^{t} \alpha_s$.

Therefore, to translate between DDPM and OT flow model inputs, we just need to:

1. **Translate the timestep:** Convert the DDPM timestep $t_{DDPM}$ into the corresponding OT flow timestep $t_{OT}$ (or vice-versa) with the same signal-to-noise ratio:

$$\frac{(1 - \sigma_{t_{OT}})}{\sigma_{t_{OT}}} = \frac{\sqrt{\bar{\alpha}_{t_{DDPM}}}}{\sqrt{1 - \bar{\alpha}_{t_{DDPM}}}}$$

2. **Rescale the input:** Rescale the noisy image by a scalar $c$ so that it has the expected magnitude:

$$c(1 - \sigma_{t_{OT}}) = \sqrt{\bar{\alpha}_{t_{DDPM}}}$$

To apply DiffC to Flux dev, we simply translate the noisy latent and timestep into DDPM format to apply the DiffC algorithm, and then back into OT flow format when performing inference with Flux.

## A.4 VISUAL RESULTS

Here are Kodak reconstructions at five different bitrates using Stable Diffusion 1.5. These images are all generated using our actual implementation of DiffC, except for the final image in Figure **??** which shows reconstructions with the theoretically ideal version of the algorithm.

## A.5 CUDA KERNEL FOR RCC

In RCC, we generate a sequence of pseduorandom samples $z \sim p$, and randomly select one with probability proportional to $w = p(z)/q(z)$. The number of samples we must evaluate increases exponentially with the encoding length, so it is important that each sample be evaluated as efficiently as possible. In our implementation, we take advantage of the fact that our distributions are isotropic Gaussians with known variance to simplify $w$ to:

$$\exp\left(\frac{\mu_q^T z}{\sigma_q^2}\right)$$

In PyTorch, we can compute the weight of each sample in the following way:

```
def compute_log_w(mu_q, K, generator, batch_size=256):
    log_ws = []
    for _ in range(K // batch_size):
        log_ws.append(torch.randn(batch_size, len(mu_q), generator=
        generator) @ mu_q)

    return torch.cat(log_ws)
```

With a custom CUDA kernel, we can avoid materializing the samples in memory, making the calculation more efficient:

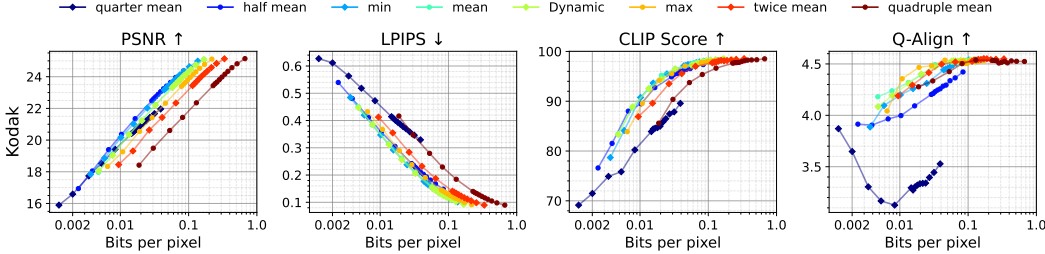

Figure 5: R-D curves for Stable Diffusion 1.5 on the Kodak dataset with various ways to determine the $D_{KL}$ per step.

```
1  __global__ void compute_log_w(
2      const float* mu_q,
3      int dim,
4      unsigned long long K,
5      unsigned long long shared_seed,
6      float* log_w
7  ) {
8      int idx = blockIdx.x * blockDim.x + threadIdx.x;
9      if (idx >= K) return;
10
11     curandState state;
12     curand_init(shared_seed, 0, idx * dim, &state);
13
14     float log_w_value = 0.0f;
15     for (int i = 0; i < dim; i++) {
16         float sample_value = curand_normal(&state);
17         log_w_value += sample_value*mu_q[i];
18     }
19
20     log_w[idx] = log_w_value;
21 }
```

### A.6    VARYING $D_{KL}$ PER STEP

At each step of the reverse channel coding process, we must communicate $q(x_{t-1}|x_t, x_0)$ using $p(x_{t-1}|x_t)$. With ideal reverse channel coding, this should require $D_{KL}(q||p)$ bits. However, both the sender and receiver must agree in advance on this $D_{KL}$ value before we can communicate the noisy sample using the PFR algorithm (Algorithm 4). This could be communicated as side information, compressed with its own entropy model. Instead, we find that these $D_{KL}$ per step values can just be hard-coded with very little impact on the performance of the DiffC algorithm. To choose the hard-coded sequence of $D_{KL}$ values, we first run DiffC for each image in our dataset while allowing $D_{kl}$ per step to be determined by the actual value of $D_{KL}(q||p)$ at each step. Then, instead of using those dynamically generated values, we fix the $D_{KL}$ values to their mean values across all images. We also try using the minimum Dkl per step across all images, and the maximum. To explore a wider range of values, we also try several multiples of the mean $D_{KL}$ schedule: one quarter the mean value, 1/2 the mean value, twice the mean value, and 4x the mean value. Figure 5 shows the R-D curves from hard-coding the $D_{KL}$ per step schedule to these different values. We find that hard-coding $D_{KL}$ values lower than the actual KL divergence results in surprisingly little distortion. A Better understanding of the impact of the $D_{KL}$ schedule on the DiffC algorithm remains an important objective for future research.

### A.7    PERFORMANCE ON OUT-OF-DISTRIBUTION IMAGE SIZES

Figure 6 demonstrates the challenge of trying to compress images which are outside of the model's training distribution. Stable Diffusion 2.1 is trained on images from 512x768 to 768x768 px in size (such as the Kodak dataset), while SDXL is trained on images around 1024x1024 pixels (such

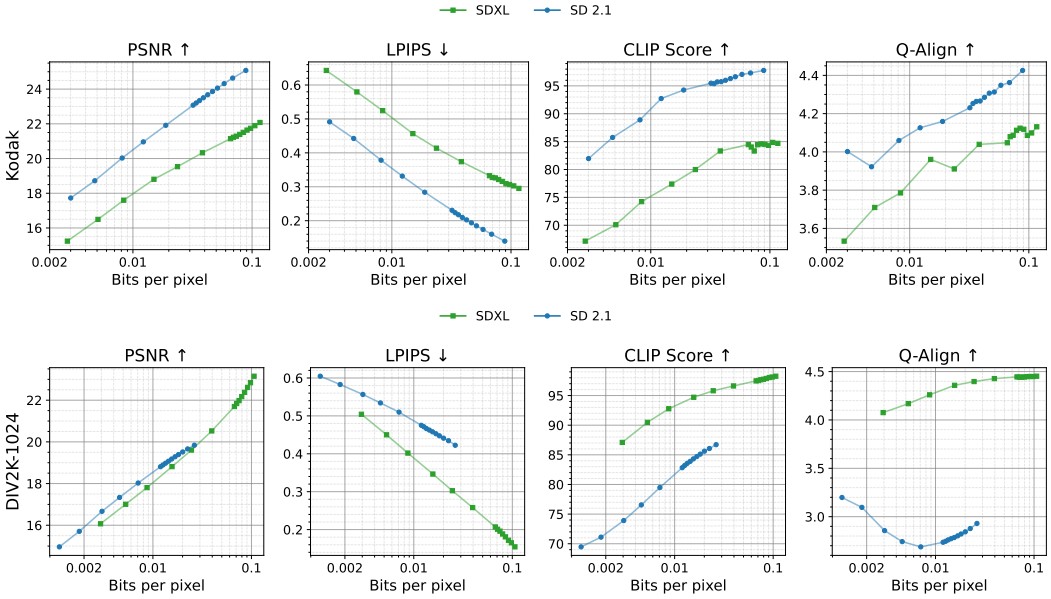

Figure 6: Rate-distortion curves for Stable Diffusion 2.1 and SDXL on Kodak vs. Div2k-1024 datasets.

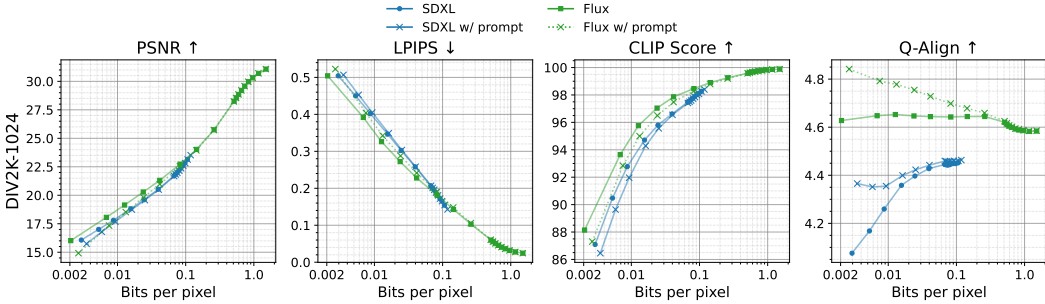

Figure 7: Rate-distortion curves for unconditional vs. prompt-guided reconstructions with SDXL and Flux. The prompt offers no improvement in distortion (PSNR, LPIPS, CLIP scores), but does considerably improve visual quality (Q-Align).

as the Div2k-1024 dataset). Rate/distortion/perception curves demonstrate a strong "home-field" advantage of these models for the resolutions they were trained for. While all other metrics follow this trend, the R-D curve for PSNR on the Div2k dataset is strangely better for SD 2.1. This is one example of a few counterintuitive PSNR results which we observe but cannot explain.

## A.8 FLUX AND SDXL R-D CURVES WITH PROMPT CONDITIONING

