# OpenReview forum: "Lossy Compression with Pretrained Diffusion Models"
_ICLR.cc/2025/Conference — ICLR 2025 Poster_

### Official Review · Reviewer_5GS8 · 2024-10-29

**Soundness:** 2
**Presentation:** 3
**Contribution:** 2
**Rating:** 5
**Confidence:** 4

**Summary:**

The paper presents a novel application of pretrained diffusion models, specifically Stable Diffusion versions 1.5, 2.1, and XL, for lossy image compression using the DiffC algorithm. The authors introduce practical workarounds to overcome challenges in reverse-channel coding, enabling efficient image compression and decompression within 10 seconds without additional training. The method is shown to be competitive with state-of-the-art compression techniques at low bitrates (0.005-0.05 bpp), demonstrating its efficacy and potential in real-world applications.

**Strengths:**

1) The paper introduces an innovative use of pretrained diffusion models for image compression, expanding the utility of these models beyond their typical generative applications.
2) The proposed method achieves competitive performance with state-of-the-art compression techniques at low bitrates, showcasing its practical value.
3) Implementation details and optimizations, such as the use of CUDA for reverse-channel coding, significantly enhance the efficiency and feasibility of the approach.
4) The publicly available implementation of the DiffC algorithm promotes transparency and facilitates further research and development in this area.

**Weaknesses:**

1) The method's performance is constrained by the fidelity limits of the Stable Diffusion models' variational autoencoders, affecting compression quality at higher bitrates.
2) The reverse-channel coding process, while optimized, still presents computational challenges, particularly for very large or very small DKL values.
3) The paper does not extensively explore the combination of DiffC with other conditional diffusion approaches, which could potentially enhance performance further.
4) The reliance on Stable Diffusion's image size limitations may affect the generalizability of the method to images outside the training distribution.

**Questions:**

1)  How does the performance of your method vary with image sizes that deviate from the training distribution of Stable Diffusion? Are there any strategies you are considering to mitigate performance degradation for such cases?
2) Can you provide a more detailed comparison between your method and other state-of-the-art compression techniques, particularly in terms of computational efficiency, quality metrics, and bitrate?
3) In the leftmost plot of Figure 3, could the authors explain why the performance of the 163-step is worse than that of the 54-step?

---

> ### Author Response · Authors · 2024-11-25
>
> Thank you for your review.
>
> In response to the raised weaknesses:
>
> > The method's performance is constrained by the fidelity limits of the Stable Diffusion models' variational autoencoders, affecting compression quality at higher bitrates.
>
> This is indeed a limitation of our method. However, note that this is much less of a limitation for our newly added results using [Flux-dev](https://huggingface.co/black-forest-labs/FLUX.1-dev), which has a much higher fidelity autoencoder.
>
> > The reverse-channel coding process, while optimized, still presents computational challenges, particularly for very large or very small DKL values.
>
> As explained in Section 3, our optimization process allows us to mostly circumvent the problem of $D_{KL}$ values which are impractically large or small. $D_{KL}$ is too small, we take larger steps. If $D_{KL}$ is too large, we break up the distribution into manageably-sized chunks. This is our main innovation on the DiffC algorithm that makes reverse-channel coding practical.
>
> > 3. The paper does not extensively explore the combination of DiffC with other conditional diffusion approaches, which could potentially enhance performance further.
> > 4. The reliance on Stable Diffusion's image size limitations may affect the generalizability of the method to images outside the training distribution.
>
> Weaknesses 3 & 4 are important points, but we regard them as outside the scope of the current work. These are improvements we plan to explore in future work.
>
> Questions:
>
> > How does the performance of your method vary with image sizes that deviate from the training distribution...?
>
> To answer your question, see the new Appendix A.7, we have added R-D curves for DiffC on OOD image sizes. While it is outside the scope of our current work, we think that DiffC could be combined with approaches like https://elasticdiffusion.github.io/ to compress images at a more flexible range of sizes and aspect ratios.
>
> > Can you provide a more detailed comparison between your method and other state-of-the-art compression techniques...?
>
> As requested, we have added comparisons to SoTA compression methods: [HiFiC](https://hific.github.io/), [PerCo](https://github.com/Nikolai10/PerCo), [MS-ILLM](https://github.com/facebookresearch/NeuralCompression/tree/main/projects/illm), and [DiffEIC](https://github.com/huai-chang/DiffEIC). As promised, our method performs competitively on rate/distortion/perception curves against these other compression methods.
>
> > In the leftmost plot of Figure 3, could the authors explain why the performance of the 163-step is worse than that of the 54-step?
>
> This is a good observation, and I wish we could explain it, but currently we cannot. Note thematically similar results in the Appendix A.7, where using lower-than-actual $D_{KL}$ values during compression can actually improve the PSNR rate-distortion curve. According to our assumptions in Section 3.2, the 163 step schedule *should* achieve the best rate-distortion curve by any reasonable distortion metric. But this conclusion depends on the assumption that our reverse-channel coding scheme will sample perfectly from the forward process. We strongly suspect that these unexpected results are downstream of some approximation error in our reverse-channel coding scheme, and some unexplored tradeoffs where it can actually be more efficient to save bits by only *approximately* instead of *exactly* sampling the forward process. We consider this an important topic in our planned future work.

---

> > ### Comment · Reviewer_5GS8 · 2024-11-27
> >
> > Thanks for your response. The author mentioned multiple times in their response that they will consider it in their future work. This should be seriously considered rather than avoided at top academic conferences.

---

### Official Review · Reviewer_2xTu · 2024-10-31

**Soundness:** 3
**Presentation:** 3
**Contribution:** 3
**Rating:** 6
**Confidence:** 4

**Summary:**

The paper applies the DiffC algorithm to pre-trained Stable Diffusion models (1.5, 2.1, and XL) for high-fidelity image compression at low bitrates. This approach leverages reverse-channel coding (RCC) to efficiently steer the denoising process toward realistic image reconstructions without additional training. By optimizing the denoising schedule and utilizing CUDA acceleration, the method achieves competitive reconstruction performance and compression times (under 10 seconds) compared to other state-of-the-art compression methods.

**Strengths:**

1. The paper extends DiffC to stable diffusion, provides an open-source implementation, and accelerates RCC with CUDA. These contributions promote the practicality of the method and pave the way for further exploration.

2. The paper is well written and easy to follow. Enough background information is provided to general readers.

3. The paper provides guidance on potential research directions in the Future Work section, offering insights for subsequent studies.

**Weaknesses:**

1. The paper lacks sufficient innovation and resembles more of an engineering improvement on existing methods.

2. The diffusion model requires multiple inference steps for both encoding and decoding, resulting in significant computational overhead.

3. Although the authors claim that their method is competitive with other state-of-the-art compression methods, no comparisons are provided in the paper. To substantiate this claim, the authors should compare their method with existing approaches such as PerCO [1], DiffEIC [2], and MS-ILLM [3] in terms of compression performance (e.g., PSNR, MS-SSIM, LPIPS, FID, etc.) and computational complexity (e.g., encoding/decoding time), presenting corresponding results in the paper.

[1] Marlene Careil, Matthew J. Muckley, Jakob Verbeek, and St{\'e}phane Lathuili{\`e}re. Towards image compression with perfect realism at ultra-low bitrates. In The Twelfth International Conference on Learning Representations, 2024

[2] Zhiyuan Li, Yanhui Zhou, Hao Wei, Chenyang Ge, and Jingwen Jiang. Towards extreme image compression with latent feature guidance and diffusion prior. IEEE Transactions on Circuits and Systems for Video Technology, 2024.

[3] Matthew J Muckley, Alaaeldin El-Nouby, Karen Ullrich, Herv{\'e} J{\'e}gou, and Jakob Verbeek. Improving statistical fidelity for neural image compression with implicit local likelihood models. In International Conference on Machine Learning, 2023.

**Questions:**

1. On page 2, line 100, the description of CDC appears to be inaccurate. CDC is trained from scratch rather than by fine-tuning conditional diffusion models.

2. In Section 4.2, why is there no R-D curve provided to illustrate the comparison between Stable Diffusion XL Base and Refiner?

3. In Section 3.5, it is mentioned, ``But in practice, we have found that just hard-coding a sequence of expected DKL values into the protocol based on their averages does not affect the performance of our method too much.`` Does this mean that the $D_{KL}$ values used in practice are manually set? Could the authors provide empirical results (e.g., R-D curves) to support the conclusion that this strategy does not affect the performance too much? Additionally, what impact would increasing or decreasing the preset $D_{KL}$ values have on performance? Could RD trade-offs be achieved by adjusting the $D_{KL}$ values?

---

> ### Author Response · Authors · 2024-11-25
>
> Thank you for your review and suggestions. Responses to your comments below:
>
> > The paper lacks sufficient innovation and resembles more of an engineering improvement on existing methods.
>
> To clarify the significance of our contribution, see both our top-level reply to all reviewers, as well as the newly added blue text in our introduction.
>
> > The diffusion model requires multiple inference steps for both encoding and decoding, resulting in significant computational overhead.
>
> This is true, and is a significant limitation of our method. While mitigating this is outside the scope of our current work, we consider it a significant opportunity for future research.
> It's worth noting that other diffusion-based generative compression methods also require multiple denoising steps. The newly added Table 1 shows that [DiffEIC](https://github.com/huai-chang/DiffEIC) and [PerCo](https://github.com/Nikolai10/PerCo) have comparable decoding times to DiffC.
>
> > Although the authors claim that their method is competitive with other state-of-the-art compression methods, no comparisons are provided in the paper.
>
> In the latest revision of the paper, we have added comparisons to [HiFiC](https://hific.github.io/), [PerCo](https://github.com/Nikolai10/PerCo), [MS-ILLM](https://github.com/facebookresearch/NeuralCompression/tree/main/projects/illm), and [DiffEIC](https://github.com/huai-chang/DiffEIC). We have also added a table with the # of model parameters and average encoding/decoding times for each method. We are pleased to report that DiffC performs competitively.
>
> > On page 2, line 100, the description of CDC appears to be inaccurate...
>
> We apologize for the mistake and have changed the wording accordingly.
>
> > In Section 4.2, why is there no R-D curve... for Stable Diffusion XL Base and Refiner?
>
> We have added an RD curve for the no-refiner version of SDXL to Figure 2. Note that it is very close in performance to the version with the refiner, but with slightly lower CLIP and quality scores.
>
> > In Section 3.5... Does this mean that the D_KL values used in practice are manually set? Could the authors provide empirical results...? Could RD trade-offs be achieved by adjusting the values?
>
> Good questions! Yes, the $D_{KL}$ values used in practice are manually set. As requested, we have added a new Appendix A.7 with further results on this topic. The figure in appendix A.7 shows the impact of increasing or decreasing the $D_{KL}$ values. RD-tradeoffs could definitely be achieved by tuning these parameters, as our figure in the appendix shows. Unfortunately, we do not yet have a good understanding of the tradeoffs involved here. We consider further analysis of this topic to be an important part of our planned future work.

---

### Official Review · Reviewer_zBvU · 2024-11-01

**Soundness:** 3
**Presentation:** 3
**Contribution:** 2
**Rating:** 6
**Confidence:** 5

**Summary:**

This paper aims to solve the reverse channel coding remaining in DiffC and to accelerate the diffusion-based compression method under 10 seconds without additional training.

**Strengths:**

Originality: The authors try their best to implement the DiffC and apply it to existing pre-trained diffusion models, such as Stable Diffusion 1.5, 2, and XL. In addition, they propose a greedy optimization technique to speed up the diffusion process and to select the best denoising timestep schedule.

Quality: The manuscript is well organized and written.

Clarity: The authors have explained their method in detail.

Significance: The significance of this work is profound because it addresses the remaining problems in the DiffC algorithm, such as inference time and reverse-channel coding.

**Weaknesses:**

(1)The manuscript looks more like a technology report than an academic paper.

(2)The authors do not provide quantitative comparisons with state-of-the-art extreme image compression methods (VQ-based methods, diffusion-based methods) and show the advantages of the proposed method.

(3) Although the main contribution of the paper is the implementation of the DiffC algorithm, the author should provide the model complexity of the proposed method (e.g., network parameters, FLOPs, encoding/decoding time, etc.).

**Questions:**

In Section 4.1, the prompts do not support image compression or reconstruction. This result contradicts many previous prompt-guided image compression methods. The author should add more analysis and explain why they do not work.

---

> ### Author Response · Authors · 2024-11-26
>
> Thank you for your review and suggestions.
>
> > The significance of this work is profound because it addresses the remaining problems in the DiffC algorithm, such as inference time and reverse-channel coding.
>
> Thank you for the kind words; we are also excited about the future of the DiffC algorithm. We hope our work helps advance the research on this algorithm from theory to practice.
>
> > (1)The manuscript looks more like a technology report than an academic paper.
>
> We have made many significant changes to the new rebuttal version of the paper, which we hope address your concerns here. See our general comment for a list of changes. New content is highlighted in blue.
>
> > (2)The authors do not provide quantitative comparisons with state-of-the-art extreme image compression methods (VQ-based methods, diffusion-based methods) and show the advantages of the proposed method.
>
> As requested, we have added comparisons to SoTA compression methods: [HiFiC](https://hific.github.io/), [PerCo](https://github.com/Nikolai10/PerCo), [MS-ILLM](https://github.com/facebookresearch/NeuralCompression/tree/main/projects/illm), and [DiffEIC](https://github.com/huai-chang/DiffEIC).
>
> > (3) Although the main contribution of the paper is the implementation of the DiffC algorithm, the author should provide the model complexity of the proposed method (e.g., network parameters, FLOPs, encoding/decoding time, etc.).
>
> We have also added Table 1, which reports the parameter counts and average encoding/decoding times for each compression method we evaluated.
>
> > In Section 4.1, the prompts do not support image compression or reconstruction. This result contradicts many previous prompt-guided image compression methods. The author should add more analysis and explain why they do not work.
>
> We were also surprised by this result. Since the last revision of the paper, we have also experimented with BLIP-generated prompts and arrived at the same conclusion. We have also added rate-distortion curves for SDXL, and Flux using BLIP prompts. While there is no clear advantage to using prompts with either SD 1.5 or 2.1, we note that both SDXL and Flux show significantly improved quality scores at low bitrates when prompts are employed (See Appendix Figure 11 on page 20).
>
> As to your question about **why** prompt conditioning is less effective for our method compared to other prompt-guided image compression methods, this is a challenging question to answer. DiffC is a different compression method with different characteristics. For now we have no concrete answer to your question; it is something we would like to understand better in our future work.

---

> > ### Comment · Reviewer_zBvU · 2024-11-27
> >
> > Thank you for your response. The prompts are more effective for generation when performing compression at extreme lower bitrates. However, when the compression ratio is small, the prompts are not effective because the pure compression network is able to generate visually pleasing reconstruction result. I have improved the score and wish you good luck.

---

### Official Review · Reviewer_kfZ4 · 2024-11-01

**Soundness:** 3
**Presentation:** 2
**Contribution:** 2
**Rating:** 5
**Confidence:** 2

**Summary:**

This submission presents a set of lossy compression method based on DiffC, with pretrained StableDiffusion1.5,2.1, and XL. To enable this utilization, a solution to the reverse-channel coding problem is proposed. This submission is also the first one to publicly release an implementation of the DiffC algorithm.

**Strengths:**

The paper implemented the SOTA stable diffusion for single image compression usage, and also released an implementation of DiffC to the public.

**Weaknesses:**

First, I don't think the main contribution is significant. The major idea had already been proposed several years ago.
Second, I failed to find the comparisons of this idea with other SOTA image compression methods.

**Questions:**

Could you present the comparisons of this idea with other SOTA image compression methods?

---

> ### Author Response · Authors · 2024-11-25
>
> Thank you for your review and suggestions.
>
> > I failed to find the comparisons of this idea with other SOTA image compression methods. Could you present the comparisons of this idea with other SOTA image compression methods?
>
> As requested, we have added comparisons to SoTA compression methods: [HiFiC](https://hific.github.io/), [PerCo](https://github.com/Nikolai10/PerCo), [MS-ILLM](https://github.com/facebookresearch/NeuralCompression/tree/main/projects/illm), and [DiffEIC](https://github.com/huai-chang/DiffEIC). As promised, our method performs competitively on rate/distortion/perception curves against these other compression methods.
>
> > First, I don't think the main contribution is significant. The major idea had already been proposed several years ago.
>
> We have also added some discussion in the introduction emphasizing the significance of our contribution. Also see our top-level reply to all reviewers for a defense of our paper's significance.
>
> Please refer to the latest revision of the paper. New content is highlighted in blue.

---

### Author Response · Authors · 2024-11-25
**General Response**

We thank the reviewers for their feedback. Because many reviewers shared two common criticisms, we will respond to those here instead of individually:

1) First, reviewers universally pointed out the lack of comparisons to other generative compression methods. We apologize for this deficiency in our original submission, and thank the reviewers for their valuable suggestions on which baselines to compare to. In the new version of the paper, you will find comparisons to [HiFiC](https://hific.github.io/), [PerCo](https://github.com/Nikolai10/PerCo), [MS-ILLM](https://github.com/facebookresearch/NeuralCompression/tree/main/projects/illm), and [DiffEIC](https://github.com/huai-chang/DiffEIC). We are happy to report that DiffC achieves competitive rate/distortion/perception curves when compared to these models.

2) The second universal concern is whether our work is sufficiently novel for an ICLR publication. We acknowledge that our work is more or less an implementation of an algorithm proposed in previous works, and that our technical contributions to that algorithm are relatively modest. However, we would like to emphasize that before our contribution, the DiffC algorithm was only a hypothetical proposal, and had never been fully implemented, due to the anticipated difficulties in reverse-channel coding. Theis et al evaluated the *hypothetical* rate-distortion curves of the DiffC algorithm assuming a *hypothetical* solution to the reverse channel coding problem. Our contributions demonstrate that reverse-channel coding is not as computationally intractable as anticipated, and allow us to provide the *first implementation* of DiffC. We believe this is a significant milestone for lossy compression with diffusion models which is worthy of research attention.

    We also think that the academic community will be interested to note that pretrained diffusion models such as Stable Diffusion and Flux are zero-shot progressive image compressors which occupy a position on the rate/distortion/perception Pareto frontier among other state-of-the-art generative compression methods. Considering that these models are not even designed with this task in mind, we think these results are very promising for the future of the DiffC algorithm.

We have added some new commentary to the introduction, highlighted in blue, to emphasize the significance of this work.

The rebuttal version of the paper includes several other significant changes, highlighted in blue:

* **We have implemented the DiffC algorithm for [Flux-dev](https://huggingface.co/black-forest-labs/FLUX.1-dev)**. To do so, we describe a way to transform optimal transport flow models into DDPMs and vice-versa, which other researchers may find useful. See section 3.4, Appendix A.3, and the results section.
* Moved the appendix on optimizing the timestep schedule into Section 3.2.
* Added a table containing the # of model parameters, encoding and decoding times for each compression method, see Table 1.
* Consolidated many rate-distortion plots into a single revamped plot; see the new Figure 2.
* Added a second figure of visual results, comparing reconstructions from DiffC, [PerCo](https://github.com/Nikolai10/PerCo), [HiFiC](https://hific.github.io/), and [MS-ILLM](https://github.com/facebookresearch/NeuralCompression/tree/main/projects/illm), see Figure 3.
* We have replaced the FID metric with [Q-align](https://github.com/Q-Future/Q-Align), a no-reference image quality metric which we believe is more meaningful than FID for our relatively small datasets (Kodak, 24 images, and Div2k, 800 images).
* Added Appendix A.2 in which we consider an alternative denoising approach. This was a negative result, but this section contains an interesting result about the correlation between diffusion latent spaces and human perception.
* Added Appendix A.6 as a continuation of our discussion about hard-coding the $D_{KL}$ values per step.
* Added appendix A7, comparing the R-D curves of SDXL vs. SD2.1 on different resolution images.

---

### Meta-Review · Area_Chair_DFS2 · 2024-12-17

**Metareview:**

The authors evaluate, to my knowledge, the first practical implementation of the DiffC algorithm.

While massaging the model to fit the constraints of the PFR algorithm does represent a pure engineering effort, the authors, to their credit, do represent it as such. I would agree with the reviewers that the novelty is somewhat weak, but I do believe the paper represents an important step for the community, showing that DiffC is feasible in practice, and giving a first idea of how well an implementation of reverse channel coding performs compared to the theory. As such it could serve as a good baseline for future work.

That said, as an engineering paper, the paper should shine with a comprehensive evaluation, which the reviewers flagged in the initial revision. The authors responded well enough. However, I'd strongly suggest that they also include a commercial method such as VVC-intra in the final paper, given that they evaluate using PSNR, among other metrics. This would widen the audience of the paper, which would be in the authors' interest.

Given the above points, I have decided to tip the perfectly balanced tie of reviews towards acceptance.

**Additional Comments On Reviewer Discussion:**

The reviewers were mostly concerned about two points: novelty and lack of comparisons to existing work. The authors did not claim that the paper is more than an engineering paper, which is fair, and added a lot more empirical results.

---

### Decision · Program_Chairs · 2025-01-22

Accept (Poster)